# Polymeric Micelles Formulation of Combretastatin Derivatives with Enhanced Solubility, Cytostatic Activity and Selectivity against Cancer Cells

**DOI:** 10.3390/pharmaceutics15061613

**Published:** 2023-05-29

**Authors:** Igor D. Zlotnikov, Alexander A. Ezhov, Artem S. Ferberg, Sergey S. Krylov, Marina N. Semenova, Victor V. Semenov, Elena V. Kudryashova

**Affiliations:** 1Faculty of Chemistry, Lomonosov Moscow State University, Leninskie Gory 1/3, 119991 Moscow, Russia; zlotnikovid@my.msu.ru (I.D.Z.); fidelche@yandex.ru (A.S.F.); 2Faculty of Physics, Lomonosov Moscow State University, Leninskie Gory 1/2, 119991 Moscow, Russia; alexander-ezhov@yandex.ru; 3N. D. Zelinsky Institute of Organic Chemistry RAS, Leninsky Prospect 47, 119991 Moscow, Russia; 4N. K. Koltzov Institute of Developmental Biology RAS, Vavilov Street 26, 119334 Moscow, Russia

**Keywords:** polymeric micelle, glutathione, combretastatin derivatives, cytometry, rhodamine 6G

## Abstract

Combretastatin derivatives is a promising class of antitumor agents, tubulin assembly inhibitors. However, due to poor solubility and insufficient selectivity to tumor cells, we believe, their therapeutic potential has not been fully realized yet. This paper describes polymeric micelles based on chitosan (a polycation that causes pH and thermosensitivity of micelles) and fatty acids (stearic, lipoic, oleic and mercaptoundecanoic), which were used as a carrier for a range of combretastatin derivatives and reference organic compounds, demonstrating otherwise impossible delivery to tumor cells, at the same time substantially reduced penetration into normal cells. Polymers containing sulfur atoms in hydrophobic tails form micelles with a zeta potential of about 30 mV, which increases to 40–45 mV when cytostatics are loaded. Polymers with tails of oleic and stearic acids form poorly charged micelles. The use of polymeric 400 nm micelles provides the dissolution of hydrophobic potential drug molecules. Micelles could significantly increase the selectivity of cytostatics against tumors, which has been shown using MTT (3-(4,5-dimethylthiazol-2-yl)-2,5-diphenyltetrazolium bromide) assay, Fourier transform infrared (FTIR) spectroscopy, flow cytometry and fluorescence microscopy. Atomic force microscopy presented the difference between the unloaded micelles and those loaded with the drug: the size of the former was 30 nm on average, while the latter had a “disc-like” shape and a size of about 450 nm. The loading of drugs into the core of micelles was confirmed by UV and fluorescence spectroscopy methods; shifts of absorption and emission maxima into the long-wavelength region by tens of nm was observed. With FTIR spectroscopy, a high interaction efficiency of micelles with the drug on cells was demonstrated, but at the same time, selective absorption was observed: micellar cytostatics penetrate into A549 cancer cells 1.5–2 times better than the simple form of the drugs. Moreover, in normal HEK293T, the penetration of the drug is reduced. The proposed mechanism for reducing the accumulation of drugs in normal cells is the adsorption of micelles on the cell surface and the preservation of cytostatics to penetrate inside the cells. At the same time, in cancer cells, due to the structural features of the micelles, they penetrate inside, merging with the membrane and releasing the drug by pH- and glutathione-sensitive mechanisms. From a methodological point of view, we have proposed a powerful approach to the observation of micelles using a flow cytometer, which, in addition, allows us to quantify the cells that have absorbed/adsorbed cytostatic fluorophore and distinguish between specific and non-specific binding. Thus, we present polymeric micelles as drug delivery systems in tumors using the example of combretastatin derivatives and model fluorophore-cytostatic rhodamine 6G.

## 1. Introduction

A large proportion of modern research in the field of medical chemistry is devoted to the development of new synthetic analogues of combretastatins with configurational stability, a strong antiproliferative effect, along with minimal side effects and the ability to overcome multidrug resistance [1,2,3,4,5]. Combretastatins consist of two phenyl rings linked by a two-carbon bridge. The compound causes mitotic arrest in cancer cells, inhibits microtubule assembly and stimulates tubulin-dependent guanosine-5′-triphosphate hydrolysis [5,6,7,8,9]. Derivatives of combretastatin—CA4P (fosbretabulin), AVE8062 and ZD6186—are at stages 2/3 of clinical trials as anticancer drugs, acting on the mechanism of destabilizing microtubules and destroying the blood vessels of tumors [10]. However, these drugs affect the cardiovascular system, cause acute pain and exhibit other serious side effects that limit their therapeutic usefulness. Significant efforts were directed to the development of new analogues of combretastatin with a limited conformation (cis), with reduced overall toxicity and improved efficiency profiles; for example, due to double bond fixation in the ring. For optimal binding to tubulin and the manifestation of high antimitotic activity, combretastatin molecules must have a cis-configuration. Five-membered heterocycles are considered as a non-isomerizable and metabolically stable bioisosteric replacement of the double bond in combretastatins, fixing the rings A and B in the cis-position. Replacement of combretastatin’s double bond with heterocyclic five-membered rings (e.g., 1,3-thiazol-2-amine, pyrazole, 1,3-thiazole, 1,2,4-triazoles, tetrazoles) does not lead to a decrease in either cytotoxic or antitubulin activities; however, it has a number of advantages: (a) prevention of combretastatin cis–trans switching; (b) increased specificity due to the absence of the trans-conformation that is non-specific for other cellular targets; and (c) possible improvement of therapeutic properties due to five-membered heterocycles [11].

In this paper, some of the most active combretastatin derivatives with five-membered linkers and tubulin polymerization inhibitors—representatives of the class of substituted arylpyrazoles, triazoles, albendazole, diarylisoxazole—are investigated in terms of their cytostatic properties. These compounds exceed the activity of natural CA4 both in vitro on 60 cancer cell lines (US Cancer Institute NCI) and in vivo in tests on sea urchin embryos [12]. Another tubulin polymerization inhibitor under study, the anthelmintic drug albendazole [13,14,15], shows comparable activity. However, all these promising molecules are poorly soluble in water, which still does not allow us to obtain solid data in higher concentrations in tests on mice, since blood lymph causes their precipitation from the studied preparative forms.

The solution to the problem of poor solubility, while at the same time increasing the selectivity of cytostatics, is the use of polymeric micelles based on chitosan, grafted with fatty acids [16,17,18,19,20,21,22,23,24,25,26,27,28,29,30,31,32]. Polymeric micelles have a radius of about 10–400 nm, which allows them to be used intravenously. Polymeric micelles based on chitosan modified with fatty acids are promising for strengthening anti-cancer and antibacterial drugs [17,18,23,24,27,29,30,32,33,34,35,36,37,38,39]. In this article, a relatively new approach is proposed using the properties of chitosan as a polycation with pKa 6.4 [40,41], ionized in such a way that, in the microenvironment of tumors, the loaded drug is quickly released by the delivery system [34,42,43,44,45,46]. In addition, chitosan can provide thermal sensitivity due to hydrophobic chain interactions at 38–42 °C. Fatty acids are necessary for the synthesis of amphiphilic polymers, forming polymer micelles which have low critical concentrations of micelle formation, are biocompatible and are highly selective. Micelles include hydrophobic substances in the core, and release the drug actively and selectively in a weakly acidic medium (pH 5.5–6) corresponding to the microenvironment of tumors. The literature describes pH-sensitive systems of drug delivery to tumors due to C=N (Schiff Base), S–S, N=N, and ester bonds, which provide high anticancer selectivity [47,48,49]. In addition, in this work, the aspect of glutathione sensitivity of micelles with S–S bonds selectively destroyed in tumor cells is realized.

A new approach based on Fourier transform infrared (FTIR) spectroscopy, allowing the nature and effectiveness of the drug’s effect on individual components of the cell to be determined with confirmation of the results by MTT tests, was applied to study the effectiveness and mechanism of the action of the micellar cytostatics formulations on tumor cells compared to normal ones. FTIR spectroscopy allows monitoring of the interaction of drugs with the main structural components of the cell (cell membrane, transmembrane and internal proteins, DNA, carbohydrates) and study of the molecular mechanism of action of drugs in a simple form and micellar; thereby achieving selectivity against cancer cells [50,51,52,53,54,55,56,57].

Rhodamine 6G [58,59] was selected as a model cytostatic, which is simultaneously a fluorophore in the visible region, to visualize the effect of micelles on the adsorption and absorption of drugs into cells using fluorescent microscopy and cytometry methods. Flow cytometry was used for the first time to prove the existence of fluorescent polymeric micelles loaded with rhodamine 6G, and the enhanced accumulation of cytostatics in cancer cells was demonstrated (micellar forms vs. free).

Thus, the study is devoted to the development of soluble forms of combretastatin derivatives and to the achievement of cytostatic selectivity against cancer cells due to stimulus (pH- and glutathione-) sensitive micelles.

## 2. Materials and Methods

### 2.1. Reagents

Chitosan oligosaccharide lactate 5 kDa, oleic acid (OA), stearic acid (SA), lipoic acid (LA), 11-mercaptoundecanoic acid (MUA), 1-ethyl-3-(3-dimethylaminopropyl) carbodiimide (EDC), N-hydroxysuccinimide (NHS), 1M 2,4,6-trinitrobenzenesulfonic acid (TNBS), and rhodamine 6G (R6G) were obtained from Sigma Aldrich (St Louis, MO, USA). Albendazole was purchased from Acros Organics (New Jersey, NJ, USA) at the highest commercial quality.

### 2.2. Synthesis of Cytostatics

Cytostatics **1**, **2**, **4** (Table 1, Figure 1) were obtained according to the techniques described by the authors earlier with ^1^H and ^13^C NMR spectra [12,60].

### 2.3. Synthesis and Characterization of Micelles

#### 2.3.1. Synthesis of Acid-Modified Chitosans

The acid-modified chitosans Chit5-SA, Chit5-OA, Chit5-MUA and Chit5-LA were synthesized by the coupling reaction of COOH of acids with the NH_2_ group using 1-ethyl-3-(3-dimethylaminopropyl) carbodiimide (EDC) and N-hydroxysuccinimide at 60 °C for 12 h (Appendix A), as described earlier [34].

Polymers were freeze-dried at –70 °C (Edwards 5, BOC Edwards, London, UK). The grafting degree was calculated according to spectrophotometric titration of acid-modified chitosans and non-modified chitosan NH_2_ groups.

FITC-labelled polymers were obtained as follows: polymer samples were incubated for 60 min at 37 °C (PBS, pH 7.4) with 1 µg/mL of FITC (in 0.1% DMSO) solution, followed by dialysis against PBS (cut-off 6–8 kDa) for 4 h.

#### 2.3.2. Preparation of Micelles

Amphiphilic chitosan-based polymers were mixed with cytostatics (1/1 *w*/*w*) in PBS (0.01 M, pH 7.4) at a concentration of 2 mg/mL. Micelle solutions were prepared by probe-type ultra-sonic treatment (50 °C, 10 min), followed by extrusion (200–400 nm membrane, Avanti Polar Lipids).

#### 2.3.3. Nanoparticle Tracking Analysis (NTA) for Determination of the Hydrodynamic Diameter of the Micelles

Hydrodynamic diameters of polymeric micelles were measured by nanoparticle tracking analysis using the Nanosight LM10-HS device (NanoSight, Amesbury, UK) and the Stokes–Einstein equation. Micelles samples were diluted with MilliQ purified water to a particle concentration of 10^9^–10^10^ particles/mL. The hydrodynamic diameter was determined by the Stokes–Einstein equation due to the analysis of the trajectory of Brownian motion of particles (Video S1). Measurements were carried out five-fold and averaged.

#### 2.3.4. Dynamic Light Scattering (DLS)

The micelles hydrodynamic diameter sizes and ζ-potentials were measured using a Zetasizer Nano S «Malvern» (Malvern, UK) (4 mW He–Ne laser, 633 nm, scattering angle 173°) in 0.01 M PBS (pH 7.4). DLS data were analyzed using «Zetasizer Software» (v. 8.02).

### 2.4. Cell Cultivation and Toxicity Assay

Adenocarcinomic human alveolar basal epithelial cells (A549 cell lines) (Manassas, VA, USA) were grown in RPMI-1640 medium. Linear cells of the embryonic kidney human epithelium (HEK293T) were cultured in DMEM (Dulbecco’s Modified Eagle’s Medium), as described earlier [34,61]. Cell viability experiments (MTT assay) were performed as described earlier [34,61].

The cell line was obtained from Lomonosov Moscow State University Depository of Live Systems Collection and Laboratory of Medical Biotechnology, Institute of Biomedical Chemistry (Moscow, Russia).

To test acute toxicity, cells were cultivated for 72 h in a 96-well plate in the presence of cytostatics (concentration range is 3 nM–0.3 mM in micelles 1:1 *w*/*w*, 3 nM–1 mM for free cytostatics), and cell viability was tested by MTT test [62,63].

### 2.5. FTIR Spectroscopy of Drug Actions on A549 and HEK293T Cells

The ATR-FTIR spectra of cell sample suspensions were recorded using a Bruker Tensor 27 spectrometer equipped with a liquid nitrogen-cooled MCT (mercury cadmium telluride) detector, as described earlier [34,56,61].

### 2.6. Fluorescence Microscopy of Cells

Fluorescence microscopy experiments were conducted using Olympus IX81 equipped by an Olympus XM10 cooled CCD monochrome camera, xenon and halogen lamps for fluorescence and brightfield imaging, and objectives UPlanSApo 20× NA 0.75 and UplanSApo 40× NA 0.90. Olympus U-MNB2 and U-MWG2 mirrors were used for blue and green excitation light, respectively (Appendix A). λ_exci_(blue) 470–490 nm, λ_exci_(green) = 510–560 nm. The fluorescence of R6G was determined at the excitation of 510–560 nm, and the fluorescence of FITC was determined by the difference between the two channels. Olympus cell Sens imaging software was used for microscope and camera control. Experimental data were analyzed using ImageJ 1.53e software.

### 2.7. Atomic Force Microscopy (AFM)

AFM experiments were conducted using a scanning probe microscope NTEGRA Prima (NT-MDT, Moscow, Russia), operated in a semi-contact mode with 15–20 nm peak-to-peak amplitude of the “free air” probe oscillations. Experimental data were analyzed using ImageJ 1.53e software.

### 2.8. NMR Spectroscopy

^1^H NMR spectra of samples (7–10 mg/mL in D_2_O) were recorded on a Bruker Avance 400 spectrometer (Germany, 400 MHz).

### 2.9. The Solubility of Cytostatics in Water Using UV Spectroscopy

UV spectra were recorded on the AmerSham Biosciences UltraSpec 2100 pro device (Amersham Biosciences, Piscataway, USA) three times. The substances were dissolved in acetonitrile, followed by UV recording of the spectra at various cytostatic concentrations, and calibration dependencies were plotted. Next, saturated solutions of substances in water were prepared and the spectra of the aqueous solutions were recorded. Considering the extinction coefficients in water and acetonitrile to be approximately equal, the solubility was determined.

### 2.10. Flow Cytometry

A CytoFLEX S flow cytometer (Beckman Coulter, California, CA, USA) was used to study micelles, as well as A549 and HEK293T cells with fluorophores (R6G). Detection of fluorescent micelles (400 nm in size) by a cytometer showed the perspective option for micelles studied with this new analytical technique. The cells were incubated with pure rhodamine 6G and rhodamine 6G in micelles (to achieve a final concentration of 5 µg/mL for each dye) for 15 or 60 min, then washed with PBS buffer and centrifuged at 2000× *g* for 5 min, followed by resuspension in PBS buffer at a concentration of 1 × 10^6^ cells/mL. The CytoFLEX S flow cytometer was set up according to the manufacturer’s instructions. The sample tube was loaded onto the flow cytometer and data were collected using the 488 nm laser for excitation. The fluorescence emissions were collected using a 585/42 nm bandpass filter. Data were collected for 10,000 cells or micelles for each sample. The collected data was then analyzed using CytExpert software. Gating was applied to exclude debris and doublets, and only single cells/micelles were analyzed. Mean fluorescence intensity (MFI) was measured for the cells/micelles stained with pure rhodamine and rhodamine in micelles. The gating was performed based on the values of front and side light scattering (high side (SSC) is typical for debris, and high front (FSC) is for cells or micelles) and the fluorescent channel (relative to cell/background autofluorescence).

### 2.11. Hemolytic and Thrombogenic Indexes of Micellar Cytostatics

The hemolytic index and thrombogenic potential of micelles were calculated using erythrocytes from human venous blood according to the previously described technique [17]. Briefly, 3 mL of human blood (voluntary donation of one of the authors of the work) was collected in tubes with 0.3 mL 2% EDTA, followed by centrifugation at 1500× *g* (4 °C) for 10 min. The erythrocytes were washed twice with 0.9% saline by centrifugation in the same conditions. The erythrocyte suspension in 3 mL 0.9% NaCl were divided into portions of 0.2 mL, followed by incubation with micelles (final concentration 0.2 mg/mL) at room temperature for 2 h. Samples were centrifuged, followed by spectrophotometric determination of released hemoglobin at 550 nm. For 100% hemolysis, 2% Triton X-100 treated samples were measured; for 0% hemolysis, 0.9% NaCl solution treated samples were used. Hemolytic index = (A550 of sample with 100% hemolysis − A550 of sample)/(A550 of sample with 100% hemolysis − A550 of sample with 0% hemolysis).

The thrombogenic potential of micelles was studied using whole human venous blood. First, 0.2 mL of 0.1% EDTA solution was added to 2 mL of blood, divided into portions and the studied micelle solutions were immediately added at the final concentration of 0.2 mg/mL. The samples were incubated at room temperature for 30 min, then 0.5 mL of distilled water was added to destroy erythrocytes not in the thrombus. The molar absorption of the samples was measured at 550 nm. For 100% thrombosis, glass microspheres treated samples were measured; for 0% thrombosis, 0.9% NaCl solution treated samples were used. Thrombogenicity index = (A550 of sample with 100% thrombosis − A550 of sample)/(A550 of sample with 100% thrombosis − A550 of sample with 0% thrombosis).

For the studied micellar cytostatics, the hemolytic index did not exceed 2% (*p* = 0.01), and the thrombogenicity index did not exceed 3% (*p* = 0.05).

### 2.12. Statistical Analysis

Statistical analysis of Cyto-tox and spectral data was carried out using the Student’s *t*-test Origin 2022 software (OriginLab Corporation). Values were presented as the mean ± SD of three or five experiments.

## 3. Results and Discussion

### 3.1. Synthesis and Characterization of Amphiphilic Polymers and Micelles

#### 3.1.1. Amphiphilic Polymers

To obtain amphiphilic polymers, chitosan polycation was chosen as the polar component, and fatty acids (SA, OA, MUA and LA) were chosen as the hydrophobic component. The synthesis was carried out by activation reactions of the carboxyl group of acids with carbodiimide, followed by crosslinking with chitosan amino groups (Appendix A)—similar to that described earlier in [61]. The variation of fatty acid residues is necessary to study the effect of the double bond (which determines the degree of fluidity of the internal hydrophobic part of the micelles), the presence of S–H groups, as well as crosslinking (–CH_2_–S–S–CH_2_–) on the properties of the resulting micelles, the degree of inclusion of drugs, and, most importantly, on the release of cytostatics in tumor cells. Selective release in tumors can be achieved due to increased glutathione sensitivity in tumor cells and the acidified microenvironment (pH sensitivity) [44].

In this paper, the molecular structure of polymers is studied using FTIR spectroscopy, and using D_2_O as a solvent; this makes it possible to analyze the state of –OH, –NH_2_ and –CH_2_ groups that are hydrated and banned in water. In the FTIR spectra of grafted chitosans (Figure 1), in comparison with non-modified Chit5, absorption bands of the initial components are observed (C–O–C oscillations of Chit5 1000–1100 cm^−1^, –NH_2_ oscillations of Chit5 3250–3500 cm^−1^ and –CH_2_– oscillations of acid residues), as well as more intense peaks corresponding to the formation of the chitosan-acid amide bond (v(C=O) 1660–1720 cm^−1^, δ(N–H) 1530–1630 cm^−1^ and v(C–N) 1200–1350 cm^−1^). ^1^H NMR spectroscopy confirms the formation of chitosan–acid conjugates (Appendix A), since chitosan peaks are present in the conjugate spectra (4.2 (H1), δ = 3.2 (H2), δ = 3.8, 4.0 (H3, H4, H5, H6, H6′), δ = 2.11 (NH–C(=O)–CH_3_)) and fatty acids (2.0–2.3, 1.25 ppm—H atoms of alkyl groups [50,51,52], 3.64 ppm—C–H near the dithiolane fragment of LA and 1.41 ppm—S–H of MUA).

#### 3.1.2. Polymeric Micelles Characterization

The formation of micelles from grafted chitosans was confirmed by AFM (atomic force microscopy), NTA (nanoparticle tracking analysis), and DLS (dynamic light scattering) methods. Using AFM, the morphology of micelles loaded with pyrazole **1** was studied in comparison with non-loaded micelles (Figure 2). The micelles loaded with the drug have a disc-like shape, 1–2 nm thick (vs. 20 nm for non-loaded micelles), and the diameter of the “disc” varied from 200 nm to 1.65 μm. The median diameter was 550 nm (SD = 150 nm). These values are strikingly different from those obtained for micelles without a drug (30 nm) and unmodified chitosan forming aggregates. In the case of micelles containing the drug, smooth dense “discs” are formed due to the compaction of the micelle structure around the drug molecules. In other words, we observe significant changes in the texture of micellar objects when the drug is turned on, which proves the effective inclusion of the drug and the formation of soluble particles. The literature also describes the use of AFM to determine the shape and morphology of lipid particles such as liposomes or micelles; when the drug is turned on, the micelles regroup and form a denser structure, since the drug initiates and compacts the micelles [28,62,63,64,65].

The existence of micelles as particles in solution was confirmed by the NTA method which registers the mobility of nanoparticles specifically (Video S1, Figure 3). Figure 3 shows distribution diagrams of micelles’ sizes formed by modified chitosan. A comparison of non-extruded and extruded samples (Figure 3a vs. Figure 3b) is presented; after passing through a membrane of 400 nm, inhomogeneous particles become beautiful micelles with orderly distribution. The inclusion of drugs (Figure 3c–e) is accompanied by additional structuring of micelles, which was demonstrated above by the AFM method.

The characteristics of micelles formed by modified chitosan are presented in Table 1. According to the NTA method, micelles have a size of about 300–400 nm; conversely, according to DLS data, the hydrodynamic size is larger, at about 400–700 nm. CMC (critical micelle concentration), determined using the Nile Red dye, as described by us earlier [61], was several orders of magnitude lower than for non-polymer surfactants [66,67,68]. The lowest CMCs are characteristic for chitosan modified with OA and MUA (5–8 nM). The zeta potential of Chit5-SA-20 particles is close to 0, which is poor from the point of view of aggregation. For the remaining micelles, the charge is positive, and also for MUA- and LA-modified chitosans of the order of +30 mV. The inclusion of drugs **1**–**5** in the micelles of Chit5-MUA-20 is accompanied by an increase in the zeta potential to 40–45 mV. This is very important for delivery to cancer cells with a negative zeta potential (about –20 mV). Indeed, when incubating cells with micelles, highly effective adsorption is observed: when increasing the zeta potential of A549 cells from –20 mV to +15 mV approximately (−10 mV for A549 with free R6G), cell recharge is observed.

### 3.2. Cytostatic Drugs Spectral Properties, Solubility and Loading into Micelles

Representatives of the class of substituted arylpyrazoles, triazoles, albendazole, diarylisoxazole, as well as the model dye and fluorophore rhodamine 6G (R6G), were selected as antitumor drugs (Table 2). However, the problem of low solubility in water even when heated is acute. Substances **1**–**5** are soluble in DMSO, but in this form they cannot be used as a drug; not only because DMSO cannot be injected intravenously, but also because crystals fall out of DMSO when diluted in a cellular environment. The extremely low solubility of cytostatics 1–4 limits its use in biomedical practice, while in micelles, the drug solubility is sufficient (Table 2).

The micrographs (Appendix A) demonstrate that cytostatics dropped out of DMSO when diluting PBS (almost instantly). The micelles contribute to the dissolution of cytostatics, and this effect depends on the concentration (Appendix A); in a shortage of micelles (5 to 1 *w*/*w*), fibers are observed instead of crystals. Complete dissolution is observed with an excess of micelles (2 to 1 *w*/*w*) after 5-min incubation—no crystals or fibers are observed in the microscope (Appendix A).

Thus, an effective approach based on the creation of micellar formulations is proposed to realize the potential of using powerful cytostatics in medicine. Organic chemists synthesize new substances, but the final step requires the creation of biocompatible formulations providing them with the improved properties.

#### 3.2.1. UV-Visible and Fluorescence Spectroscopy

The inclusion of drugs into the micelle core should be accompanied by a change in the hydrophobicity of the fluorophore microenvironment, which is reflected in the electronic spectra. When the micelle is incorporated into the hydrophobic core, the microenvironment of aromatic systems of cytostatics **1**–**5** changes, which is reflected in the absorption and emission spectra of fluorescence (Figure 4 and Appendix A). Figure 4a shows the absorption spectra of diarylisoxazole **4** in a water-acetonitrile mixture (90/10 *v*/*v*) and in the micellar system. The micellar form is characterized by a peak consisting of two components: hydrophilic (water and polar chains of chitosan) and hydrophobic (fatty tails in the core of micelles). Taking into account the integral proportions of the components, the distribution of cytostatic substances in the core/coat is approximately 60/40. The obtained result is also confirmed by fluorescence spectroscopy. Figure 4b and Appendix A show the emission spectra of cytostatics fluorescence: when incorporated into micelles, a shift of the maxima to the long-wavelength region up to 20 nm occurs, which confirms an increase in the hydrophobicity of the micro-environment of fluorophores. For R6G, this effect is the most pronounced. The total load of cytostatics (drug in the core of micelles and drug interacting with chitosan chains) exceeds 85–90%; that is, less than 15% remains unloaded. The solubility of cytostatics in micelles is about 0.05–0.1 mg/mL.

#### 3.2.2. FTIR Study

FTIR spectroscopy allows us to study the molecular details of the interaction of drugs with micelles, to find out which functional groups are involved. Figure 5 and Appendix A show the FTIR spectra of micelles from amphiphilic polymers non-loaded and loaded with drugs **1**–**5**. After loading cytostatics into micelles, characteristic bands increase in intensity in the FTIR spectra, corresponding to the oscillations of C=C, N=N, C=N in aromatic systems (1520–1650 cm^−1^), C–O–C of CH_3_–O–aryl (1270–1340 cm^−1^), and CH_3_–O (2835 cm^−1^). In the FTIR spectrum of rhodamine 6G, when included in the micelles, there is an increase in peak intensity by 1700–1750 cm^−1^, corresponding to C=O of the ester (Figure 5b). The formation of micelles is confirmed by a decrease in the intensity of the N–H peak of Chit5 (3300–3500 cm^−1^), which indicates the structuring of chitosan polar chains from aggregates into ordered assembles. The compactification of micelles and core formation when the drug is incorporated in the micellar structure is confirmed by the shift of the peaks of the CH_2_ groups of acyl tails of micelles (3000–2800 cm^−1^) into the low-frequency region.

FTIR spectroscopy provides valuable information about the micelles structure when the drug is incorporated, but the use of complementary methods will allow a better analysis of the process. NMR spectroscopy is necessary to determine the state of individual atoms; for example, –SH groups that provide sensitivity to glutathione, which is in excess in tumors. Figure 6 shows the ^1^H NMR spectra of rhodamine 6G loaded into micellar systems. Chemical shifts corresponding to chitosan protons of 3.25–4.2 ppm and protons of the aromatic system R6G 6.3–8.4 ppm are detected in the spectrum. Chit5-MUA-20 polymers form disulfide bonds (S–S) upon oxidation with glutathione, which is reflected in the NMR spectrum: the intensity of the peak of 1.4 ppm (S–H) decreases and peaks appear in the region of 2.3–3.1 ppm (–CH_2_–S–S–CH_2_–). The formation of S–S crosslinked polymers gives the Ox/Red system sensitivity, which is especially important for targeted delivery to tumors, where the drug will be released when the S–S bonds break.

Appendix A shows ^1^H NMR of Chit5-OA-20 non-loaded and loaded with pyrazole **1** and triazole **2**. In the spectra of micelles with the drug, the intensity of the peaks of the CH_2_ groups decreases (1.2–1.4 ppm) and the group of peaks 3.6–3.7 ppm (H3–H6 of Chit5) overlaps with the peaks of the drug’s methoxy groups. Thus, NMR spectroscopy confirms the synthesis of amphiphilic polymers of chitosan and fatty acids, and the micelle’s formation with the incorporated drug.

### 3.3. Selectivity of Cytostatics in Relation to Cancer or Normal Cells Using FTIR Spectroscopy

FTIR spectroscopy is a powerful method for determining the effect of a drug on cells (the state of functional groups and in the main components of cells), since we have recently developed an effective approach that makes it possible to monitor the effectiveness of both cytostatics on tumor cells and antibacterials on bacteria, and we can observe selectivity against tumor cells and a protective effect on normal cells [56]. FTIR spectroscopy is a sensitive method that allows the monitoring of changes in the main components of cells (lipid membrane 2850–3000 cm^−1^, proteins 1500–1720 cm^−1^, DNA 1240 cm^−1^, carbohydrates 1000–1100 cm^−1^) when interacting with the drug. Figure 7a shows the FTIR spectra of A549 cancer cells after 12 h of incubation with cytostatics and micelles. A decrease in the intensity of peaks relative to the control indicates a decrease in the number of viable cells and a change in the state of the living (become dying). The intensity of the peaks of amide I (1600–1700 cm^−1^) and amide II (1500–1600 cm^−1^) correlates with the number of viable cells (Figure 7b), which is shown with the use of the dye trypan blue (coloring mainly dead cells).

Micelles significantly enhanced cytostatic activity against cancer cells, but it is important that the micellar drug does not destroy normal cells. Figure 7c shows the FTIR spectra of normal HEK293T cells after 15–60 min of incubation with rhodamine 6G. The area of 1530–1600, as well as 1250–1500 cm^−1^, characterizes the amount of R6G adsorbed or absorbed by cells. After 15 min of incubation, micellar rhodamine is much higher than free R6G adsorbed on cells; however, penetration of the drug into normal cells from micelles is difficult, which is confirmed by a slight difference in the green and blue spectra (Figure 7c). In other words, the dye is stuck in the form of micelles and weakly penetrates in HEK293T. At the same time, free R6G adsorbs more slowly, but penetrates well into normal cells, which is reflected in a dramatic change in the spectrum of 15 min vs. 60 min of incubation: red and pink (Figure 7c).

Thus, selectivity against cancer cells and a protective effect on normal cells when using micellar forms of cytostatics are demonstrated.

### 3.4. MTT Assay of Anti-A549 Activity of Cytostatics

To confirm the validity and correctness of the interpretation of the FTIR spectroscopy data, we use the direct method—an MTT study, a quantitative criterion for the effectiveness of antitumor drugs. Chit5-MUA-20 cross-linked S–S micelles with the property of glutathione sensitivity and selective drug release in tumors showed the best antitumor activity; the main experiments were carried out with them. Figure 8 shows the curves of cell survival dependence on the concentration of cytostatics in DMSO in comparison with those in micellar form. IC50 was almost an order of magnitude lower in micelles than for free derivatives of combretastatin and cytostatics (Table 1). Therefore, micellar systems with cytostatics are potential anti-cancer drugs. The synthesized micellar formulations have a high antitumor potential, surpassing known cytostatics (cisplatin [69,70,71], doxorubicin [72]) and comparable to paclitaxel [73]. All the studied drugs turned out to be approximately equally effective, but the selling albendazole showed the worst results, which means that our cytostatics are promising in aspects of anticancer therapy.

Micelles selectively act against cancer cells, while we have previously shown that micelles have a protective effect on normal HEK293T cells due to differences in morphology and membrane. Presumably, micelles restore the membranes of normal cells, and penetrate into cancer cells within membrane defects. Fluorescence microscopy should be used to study the mechanisms of action of micelles, as well as the localization and duration of action.

### 3.5. Optical Microscopy Visualization of Drug Absorption

Visualization of drug penetration into cells using fluorescence microscopy is an important aspect complementing the methods of MTT and FTIR spectroscopy. The main task of fluorescence microscopy is to study the effect of micellar and simple R6G on cells, to find out how the cytostatic is adsorbed and penetrates over time and whether the micelles themselves penetrate into the cells. Figure 9 shows the lumen and fluorescent images of A549 cells in the red channel (R6G), and one of them shows a channel of FITC-labeled micelles for the micellar series in comparison with the free rhodamine itself. The intracellular concentration of rhodamine 6G increases in a series of 15–30–60 min of incubation; while, on the contrary, decreases by 24 h. After 24 h, the number of cells decreases significantly, and their internal structure is disrupted for all studied systems. If the drug accumulated in the cytoplasm and nucleoli before 1 h of incubation, then, by 24 h, a continuous weak color of “half-dead” cells is visible.

One interesting aspect is the fate of micelles, and whether they penetrate into cancer cells or not. FITC covalently attached to the micelles by the chitosan amino group shows the localization of micelles inside the cells at times up to 1 h of incubation; after 24 h, the micelles with the drug are distributed throughout the cell volume. In other words, in an hour, the micelles penetrate into the cell and accumulate in the area of the nucleus, and in 24 h, they distribute evenly throughout the cell due to the destruction of the internal structure.

Consider the colocalization of the drug (R6G), micelles and cell nuclei by 4′,6-diamidino-2-phenylindole (DAPI), on microphotographs of the overlay of three corresponding fluorescent channels (Figure 10). In the case of micellar R6G, a high degree of colocalization of the rhodamine channel, FITC channel (micelles) and DAPI is observed, which indicates an effective accumulation of R6G in cells due to absorption of micelles by A549 cells.

On the contrary, free rhodamine 6G is distributed mainly in the extracellular medium, since it is released during efflux (it is not colocalized on the DAPI channel). To correlate cell survival with the permeability of the drug (depending on the formulation) and the role of micelles in this aspect, DAPI-related experiments were carried out. Figure 11 shows the cell-associated fluorescence of R6G and FITC, which characterize the effectiveness of the drug adsorption and permeability, as well as DAPI (at short dye exposure times of 5 min), associated mainly with dead cells. Micelles increase the accumulation of rhodamine inside A549, depending on the type of micelles and the time of observation. An increase in cell-associated fluorescence of DAPI indicates an increase in the number of dead cells: the difference between simple rhodamine and micellar is almost two-fold. The greatest effect was shown for S–S stitched micelles, selectively releasing the drug in the tumor cells: cell-associated fluorescence increased by more than two-foldin comparison with free R6G. This effect of Red-Ox micelles on the effectiveness of rhodamine for durations of 30 min and more is explained by an excess of glutathione in tumor cells and the destruction of disulfide bonds inside cells (micelles with the drug penetrate together) with the simultaneous release of the drug.

### 3.6. Flow Cytometry Assay

#### 3.6.1. The Study of the Micelles as Individual Particles

The study of the micelles as individual particles was carried out using flow cytometry where we observed the micelles formation, as well as the interaction of micellar and simple forms of the drug with the cells. Flow cytometry allows the identification of objects with a size of more than 100–200 nm. Figure 12a shows diagrams of the distribution of the particles (micelles) by the intensity of R6G fluorescence. The cytometer detects micellar particles, which means they formed as particles and are loaded with a drug–fluorophore, and exhibit different fluorescence properties depending on the micelles composition. This indicates that flow cytometry is sensitive to detecting the formation and differentiation of the micellar particles.

#### 3.6.2. The Effect of Micellar Drugs on Cells

The flow cytometry method was applied to the study of micelles and their interaction with A459 and HEK293 cells. The analytically significant populations are the P2 and P3 fractions (Figure 12b,c), which correlate with the number of living and dead cells, respectively. Changes in them correspond to the effectiveness of cytostatics. P2 and P3 assignment was determined relatively to positive and negative control: intact cells (negative control) and cells with 24-h incubation with rhodamine (limit case—positive control) were selected as reference points. Figure 12b,c show that, after 15 min of incubation with rhodamine 6G (free or in micelles), mostly living A459 cells are observed (Figure 12b; green and red lines). After 1h exposition with micellar R6G, the splitting of the cell population into damaged cells and intact cells is clearly observed (purple lines in Figure 12b). For comparison, Figure 12c shows the location of the fraction for live cells (red line), damaged cells (purple line) (after 24 h of R6G exposition) and non-colored cells as a control (autofluorescence—green line). As such, the comparison of Figure 12b,c shows that, after 1 h incubation of A459 cells with micellar rhodamine, a pronounced division of the cell population into intact and damaged cells is observed.

Based on these P2 and P3 fraction assignments, more detailed information on the effect of the drug in micellar systems on the cell’s state depending on the drug exposure time was obtained for both types of cells, tumor or normal. According to cytometry data (Figure 13), micellar R6G is better adsorbed on the surface of A549 and better penetrates into cancer cells due to the creation of defects in the membrane, so the percentage of cells in the P2 population (correlating with the number of living cells) decreases (Figure 13). Micelles with MUA (S–S stitched) are more effective than those based on OA, since the former are glutathione-sensitive, and quickly recharge cells and release rhodamine due to a positive charge +30 mV (Table 1), which is reflected by the percentage of cells in the P2 population (60% vs. 40% for OA and MUA micelles, respectively). Thus, cytometry confirms the above data on the higher efficiency of the micellar form of cytostatics.

A different picture is observed in the case of HEK293T cells (Figure 14). Flow cytometry histograms of HEK293T cells incubated with R6G in both simple and micellar forms are shown in Figure 14. Free R6G adsorbs quickly on HEK293T cells in 15 min. There is a splitting of R6G+ cells into two sub-populations (Figure 12d; live on the right and partially damaged on the left). Greater permeability and a higher percentage of damaged cells, respectively, is observed at 60 min incubation. Micellar R6G at 15 min incubation is mainly adsorbed on the cell surface (the observed peak is narrow and homogeneous—Figure 12d). After 60 min exposure of micellar rhodamine with HEK293T cells, there is a broadening of the peak, which corresponds to an increase in R6G permeability.

To find the correlation of permeability and the ratio of living–damaged cells, an experiment with DAPI was conducted (Figure 15). DAPI marker penetrates mainly into dead cells after R6G treatment. Thus, with flow cytometry, it is possible to study the effect of a cytostatic–fluorophore on cells by observing the DAPI fluorescence signal intensity.

Figure 15 shows histograms of flow cytometry of HEK293T cells incubated with micellar cytostatics (pyrazole 1 and triazole 2) and control R6G, followed by DAPI staining to determine the dead/living cells ratio. The green population correlates with the number of dead cells (relative to the control R6G–). Histograms of flow cytometry of HEK293T with rhodamine 6G and DAPI clearly demonstrate an increase in the proportion of dead cells during incubation of HEK293T cells with free rhodamine. A radically different situation is observed for micellar systems: the number of damaged cells (DAPI+) is noticeably lower after 15- and 60-min incubation.

A similar situation is observed for cytostatics 1 and 2 in micelles; they practically do not inhibit HEK293T growth (there are no significant differences from the control of R6G–). Conversely, non-micellar cytostatics have pronounced cytotoxic effects on HEK293T, since the population P1 decreased. Thus, the direct method based on DAPI fluorescence (which dyes only damaged cells) demonstrated that micellar systems have a protective effect on normal cells (HEK-293) compared to free drug molecules, both in the case of model rhodamine and for cytostatics of the combretastatin derivatives series.

The data on DAPI absorption correlate well with those obtained by fluorescence microscopy, FTIR and MTT tests.

Thus, with flow cytometry, the existence of fluorescent micelles has been confirmed, which protect normal cells, but activate the actions of cytostatics against tumor cells due to greater permeability of micellar particles.

## 4. Conclusions

Combretastatin derivatives (tubulin polymerization inhibitors) are promising from the point of view of cytostatic action potential, but proved to be inapplicable due to their extremely low solubility and tendency to form crystals (even after dissolution in DMSO) when diluted in a culture medium. Polymeric chitosan micelles made it possible to solve combretastatin derivatives and practically increase the solubility by two orders of magnitude—from 1 mg/L to 0.1 mg/mL.

An important aspect of delivery systems is to achieve selectivity of action against tumors; otherwise, drugs bring more side effects than targeted treatment. The selectivity of the presented micellar cytostatics against cancer cells is realized due to the pH thermosensitivity (described by the authors earlier), and also the new property of glutathione sensitivity of micelles (S–S stitched polymers) presented here. Disulfide bonds in micelles are destroyed in tumors where there is an excessive amount of glutathione, as well as in the microenvironment of tumors; the paper implements this approach using example micelles with disulfide bonds.

The cytostatic activity of combretastatin derivative formulations against A549 cancer cells exceeds the activity of classical cytostatics used in medical practice (doxorubicin, paclitaxel, cisplatin, vincristine) by 1–2 orders of magnitude. At the same time, the developed micellar formulations are less harmful to the body, which was shown using healthy HEK293T cells.

We have proposed original techniques which allow the study mechanisms to be studied from a new point of view that previously could not be described clearly enough. The MTT test data are confirmed by FTIR spectral data on the interaction of cytostatics with the cell membrane, proteins and DNA. Using FTIR spectroscopy, it was shown that micelles act as drug enhancers on tumor cells due to pH and thermal sensitivity to the tumor microenvironment, as well as having a protective effect on normal cells (there are practically no changes in the FTIR spectra). Using CLSM, the selectivity of cytostatics against tumors was visualized, and the distribution of the micelles themselves in cells was demonstrated, which is well coordinated with the DAPI signal from the nuclei.

For the first time, the formation of micelles was proved using flow cytometry, and their selectivity of action on tumor cells using the model fluorophore R6G was also shown. The advantage of flow cytometry is the possibility of observing several fractions of particles (distinguishable by light scattering and fluorescence), which makes it possible to study the dynamics of the adsorption and penetration of the drug into the cells. An important property of micelles turned out to be the positive charge (ζ-potential is 30 mV), providing efficient adsorption even with the recharging of the cancer cells surface, and further active penetration of the micellar drug into the tumor cells, while permeability of the normal cells for the micellar formulations is much lower. This result is confirmed using CLSM, where the penetration of drugs with micelles into the cancer cells over time was visualized with DAPI (nucleus)—FITC (micelles)—R6G (drug) colocalization; in an hour, the micelles penetrated into the cell and accumulated in the area of the nucleus, and in 24 h, distributed evenly throughout the cell due to the destruction of the internal structure.

Thus, polymeric micelles have huge potential in terms of the possibility of using new drugs (having problems with solubility and/or toxicity) with the function of targeting cancer cells.

## Data Availability

The data presented in this study are available in the main text and Appendix A.

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
