# Peer review of "Polymeric Micelles Formulation of Combretastatin Derivatives with Enhanced Solubility, Cytostatic Activity and Selectivity against Cancer Cells"

_pharmaceutics, 2023, doi:10.3390/pharmaceutics15061613_

Round 1
Reviewer 1 Report
The authors here aim to present the use of polymeric micelles for cancer delivery by encapsulating combretastatin derivatives.
There are a number of questions I have:
1. There is no proof that these particles are micelles-there is no f value to substantiate their claim so it is incorrect to label these particles as micelles. There is no CAC (critical aggregation concentration) data of these materials either.
2. DLS and zeta potential data are lacking
3. There is no GPC data of the polymers
4. The study on the cell lines is highly unscientific, they chose A549 cells- a lung cancer cell line derivative to prove their IC50 values were lesser than the free form (Pg 15 L366) and yet they chose HEK29T cells- a kidney cell to prove that the carriers were not toxic. This has no logic. A healthy non-cancerous lung cell should be used as a control
5. Cytox data for IC50 does not show the log/sigmoidal form so maybe it is better to plot them as bar graphs
6. Cellular uptake data has no DAPI staining so it is unclear as to whether the cells are dead or alive. Seems to be a lot of nonspecific staining in almost all samples. Bright field images are unclear.
7. Table 2 and Figure 8 seem to be inconsistent-As per Table 2 for R6G in FITC labelled Chit5- MUA20 there is apparently no uptake in 15, 30 and 60 minutes and at the 24 h time point all particles seem to have been incorporated into the cells. However, Fig.8 for the same formulation (green bar with diagonal stripes) shows highest uptake at 30 mins
8. References are inadequate
This study has serious flaws and cannot be accepted in its current form
Grammatical flaws, long winding sentences and poor punctuation often makes the reader lose track and the essence of what was trying to be conveyed is completely lost
Reviewer 2 Report
In this manuscript, the authors modified chitosan polymers with fatty acids and use them to prepare polymeric micelles. The hydrophobic drugs were loaded to the micelles, improving their solubility and achieving the cytostatic selectivity against cancer cells. However, the authors did not explain the novelty of this manuscript compared with previously reported references. Additionally, the layout of figures and tables is messy and the quality of the English is poor. As the authors did not present any scientific insight on the field, I cannot recommend it for publication in current form.
The quality of the English is poor. There are many spelling and grammar errors, please spend more time to revise them.
Reviewer 3 Report
Dear Authors
The article entitled “Polymeric micelles as stimulus-sensitive systems for dissolving of combretastatin derivatives and their cytostatic activity against cancer cells ” is a very interesting work. Fallowing are few suggestions which might be helpful for the improvement of the article a bit
1. What is the specific significance of five-membered heterocycles in cancer treatment?
2. What is the reason for choosing specific cationic chitosan over other options polymers, and what function do fatty acids play in the development of micelles to target cancer cells?
3. For a better understanding of the research, it is best to explain the synthesis of Cytostatics 1-4 in a nutshell.
4. Along with the FTIR data, provide the NMR spectra to support in the formation of the Chit5-SA, Chit5-OA, Chit5-MUA, and Chit5-LA.
5. In section 2.4, authors should provide the concentration range of the micelles formulation for the MTT experiment.
6. In section 2.5, authors should provide brief descriptions of the FTIR spectroscopy investigation of pharmacological activities on A549 and HEK293T cells.
7. “The excitation and emission wavelength ranges were selected by Olympus U-MNB2 and U-MWG2 fluorescence mirror units for blue and green excitation light respectively”. Section 2.6 requires the wavelength.
8. The cytostatics' water solubility should be tested only after synthesis. As a result, it is recommended that section 2.9 of the method section should rearrange.
9. What is the rationale behind providing just the particle size distribution of NTA Chit5-OA-20 loaded with pyrazole but not other polymer formulations? For a better understanding, authors might display all the formulations' particle size distribution in a tabular format.
10. What is the significance of referring to the drug-loaded micelles as "pancakes"?
11. Authors should give supporting evidence for their claim of significant changes in the texture of micellar objects when the drug is loaded. Why do the NTA data and AFM data differ considerably?
12. Micelles are self-assembling nanostructures with the most frequent particle size of <200nm. However, in NTA and AFM data authors indicated that the particle distribution was 31-363nm (NTA) or 200 nm to 1.65 μm . The author should explain their findings and give evidence to support them.
13. Why do the authors just look at AFM for Pyrazole 1 loaded micelles and not others? What is the relevance of the non-loaded Chit5-MUA-20 AFM data?
14. Why did authors not include Albendazole and Diarylisoxazole in Chit5-OA-20?
15. On what grounds do authors choose Chit5-fatty acid for various Cytostatic drugs? Chit5-OA-20 is just for Pyrazole and Triazole, Chit5-MUA-20 is for albendazole, and Chit5-LA-20 is for diarylisoxazole.
16. Table 2 should be changed as a figure legend.
17. To support the claim of pH, temperatures, and glutathione sensitivity of micelles, provide in vitro release data of drug-loaded micelles in different pH and temperature conditions.
18. Although FTIR played an important part in this study, the writers failed to convey its relevance in the introductory section. The involvement of the FTIR in this investigation might be included by the authors to claim a special novelty of their work.
19. Conclusion should be revised to provide the particular and significant conclusion of the research.
The English language in the introduction and conclusion was quite poor and required extensive improvement.
Round 2
Reviewer 1 Report
The cytotoxicity data is presented on two different cell lines as per the logic presented "both types of these cells are epithelial and therefore the comparison is correct. Cells are used because they are bred to make it convenient to work: they are genetically stable (to make the experiments reproducible), they grow in a reasonable time, absorb drugs in a reasonable time, these are immortalized cells (growing indefinitely). Real healthy cells cannot be studied in such conditions – they have limited growth."
" is not scientific. In order to prove the safety of any new material a healthy (non cancerous cell line) must be chosen (HDFn-easy to grow with a doubling time of ~32 hours).
The safety of the materials cannot be proved using the experiments and data presented. This data must either be presented or a a discussion added where the safety of the particles is addressed.
Further, the IC50 curves have still not been presented and the use of R6g labelled "micelles" in the new fgure do not shed any light as to whether there is non specific staining as proper controls are lacking. The data must be presented in a form where the panels clearly show uptake of the labelled micelles, a DAPI and a merge to prove the uptake of the dye labelled particles within the cells.
Fig 9 , the two pink bar graphs look almost identical in color, please change one so that it is easier to distinguish.
The following references must be added:
https://www.sciencedirect.com/science/article/abs/pii/S092777651830763X
https://onlinelibrary.wiley.com/doi/abs/10.1002/anie.200301694
https://pubs.acs.org/doi/10.1021/acs.molpharmaceut.0c00247
Please do a grammar check to correct the syntax and phrasing of sentences along with a spell check
Reviewer 2 Report
The authors have carefully revised the manuscript. I think it should be OK for publication.
OK
Author Response
Dear Reviewer ! The authors are very grateful to you for your positive feedback on the article.
Reviewer 3 Report
Dear authors
The paper has been thoroughly revised and properly planned.
A spelling check is needed
Author Response

(The authors gave the same response as above.)
